# Physical Activity and Sedentary Behavior Assessment: A Laboratory-Based Evaluation of Agreement between Commonly Used ActiGraph and Omron Accelerometers

**DOI:** 10.3390/ijerph16173126

**Published:** 2019-08-28

**Authors:** Shohei Yano, Mohammad Javad Koohsari, Ai Shibata, Kaori Ishii, Levi Frehlich, Gavin R. McCormack, Koichiro Oka

**Affiliations:** 1Institute for Sport Sciences, Waseda University, Saitama 359-1192, Japan; 2Faculty of Sport Sciences, Waseda University, Saitama 359-1192, Japan; 3Behavioural Epidemiology Laboratory, Baker Heart and Diabetes Institute, Melbourne, VIC 3004, Australia; 4Melbourne School of Population and Global Health, The University of Melbourne, Melbourne, VIC 3010, Australia; 5Faculty of Health and Sports Sciences, University of Tsukuba, Ibaraki 305-8574, Japan; 6Department of Community Health Sciences, Cumming School of Medicine, University of Calgary, Calgary, AB T2N 4Z6, Canada

**Keywords:** objective assessment, accelerometer, activity monitor, physical activity, measurement

## Abstract

Different models of accelerometer have the potential to provide a different estimate of the same physical activity or sedentary behavior. Our study compared the outputs of the Active Style Pro (ASP) and ActiGraph (AG) devices in assessing predicted metabolic equivalents (METs) for specific activities under laboratory conditions. Thirty healthy young adults wore two hip accelerometers (ASP and AG), simultaneously while performing twenty-two activities (eight sedentary, eight household, and six ambulatory activities) in a controlled laboratory setting. For the AG, predicted METs for each activity was calculated using four equations based on vertical-axis and vector magnitude data. Separate paired *t*-tests and Bland–Altman analysis examined the difference and agreement in METs between AG using four commonly used equations and ASP measurements for each activity. AG devices using different equations calculated significantly different outcomes for most activities compared with ASP devices. The smallest differences in predicted METs estimates between ASP and AG were observed for ambulatory activities. Ambulatory activities demonstrated the best agreement between ASP and AG regardless of which AG equation was used. Our findings can be used to assist researchers in their selection of accelerometer and output estimation equations for measuring physical activity and sedentary behavior in adults.

## 1. Introduction

Accelerometers can objectively measure different aspects of physical activity and sedentary behavior including frequency, duration, intensity, and volume [1]. To date, various models of accelerometer have been used to capture physical activity and sedentary behavior in epidemiological research [2,3,4]. However, several studies have shown disagreement in estimates of the same physical activity and sedentary behavior calculated by different types of accelerometer [5,6,7,8]. Different physical activity and sedentary behavior outputs between accelerometers may be caused by device design and hardware, algorithms, settings used to estimate behavior (i.e., energy expenditure (EE) and categorized physical activity intensity), and the extraction or cut point methods used for analyzing raw outputs such as vertical axis (VT) data and vector magnitude (VM) data [9,10,11,12]. Additionally, different estimates of the same physical activity and sedentary behavior can occur among different generations or models of accelerometers within the same brand despite applying the same algorithm and settings [13,14,15]. Understanding and testing physical activity and sedentary behavior output differences between accelerometer devices is important for providing a consistent interpretation of previous epidemiological studies that have used different accelerometers.

ActiGraph (AG) accelerometers are the most-commonly used triaxial accelerometers (ActiGraph, Pensacola, FL, USA) [16]. The AG provides a high degree of freedom for researcher and user processing activities (i.e., modification of epoch length, cut-points, equations, placement, and analysis). Epoch involves the raw device data summed to a user-specified time, while cut-points and equations are used in the analysis or processing. The AG measures acceleration using three individual axes (VT, antero-posterior axis, and medio-lateral axis) and uses counts which are converted from their acceleration in each axis and VM of these three axes as raw data. The equations and cut-points were developed based on methods processing this acceleration to converts epoch count into estimates of EE and/or physical activity intensity (i.e., sedentary behavior, light intensity of physical activity, and moderate-to-vigorous physical activity [17]. Notably, the AG also has several different equations for estimating EE from the accelerometer raw data (count). Despite there being various types of equation for estimating EE for ambulatory activities (i.e., walking and running on the treadmill) and lifestyle activities, the Freedson et al. [17] equation is the most commonly used. The Freedson equation collects data for VT based on ambulatory activities at three different speeds on the treadmill in young adults [17]. Likewise, the Sasaki et al. [18] equation is also commonly for estimating EE, incorporating VM data from all three axes (vertical, antero-posterior, and medial-lateral). The Sasaki equation is also based on ambulatory activities at four different speeds on the treadmill in young adults [18]. Further, the Santos-Lozano et al. [19] equations were obtained for the VT (Santos-Lozano VT) and the VM (Santos-Lozano VM), and factored in the users age to suit their physical activity exposure in specific age groups. Santos-Lozano VT and VM equations were developed using four ambulatory activities (treadmill at four different speeds), resting, and sit-to-stand [19]. Given that these equations use slightly different combinations of input data from the accelerometers (e.g., VT, VM), it is expected that EE estimated even for the same activities may differ, however, the extent of these differences remain unknown.

A recently developed triaxial accelerometer, Active Style Pro (ASP; Omron Healthcare, Kyoto, Japan), has been used in several epidemiological studies [20,21,22]. The ASP has a proprietary algorithm, which has been promoted by the manufacturer to distinguish among household activities and locomotive activities using the ratio of filtered to unfiltered acceleration from three directions [23]. Each of the three signals from the triaxial accelerometer ASP passes through a specific filter to remove the effects of gravitational acceleration. Despite various challenges pertaining to the measurement accuracy of measuring light intensity of physical activity, especially for household activities [24], the ASP has more than a 95% classification percentage for detecting household and locomotive activities. Furthermore, the ASP adapts a multiple equation model (i.e., household and locomotive equation) to accurately estimate metabolic equivalents (METs) after discriminating the household and locomotive activities. Multiple equation models of the ASP have shown good agreement for estimating METs between the ASP and Douglas bag method [25]. In addition to its adequate measurement properties, the ASP has other features that in some circumstances might be useful for researchers. For example, the ASP calculates METs without the need for the researcher to decide which algorithm or cut-point is used. This design feature of the ASP, which limits processing decisions (i.e., modification of epoch length, cut-points, equations, placement, and analysis) may result in reproducible and therefore comparable estimates physical activity and sedentary behavior between studies which use the ASP. Furthermore, the ASP has a liquid crystal screen, which can show outputs of the activity data such as activity intensity (METs) and steps in real-time. Thus, in health promotion interventions, ASP accelerometer has the potential to promote physical activity providing the wearer with real-time feedback [26]. The display setting can also be turned off should the researcher require the wearer be blinded from their activity levels. Thus, the ASP can be a useful device for investigating the effect of self-monitoring on physical activity and can be used in intervention studies. However, since the AG and ASP are developed by different commercial companies (with different algorithms, settings, and researcher controllability), they may provide different estimates of the same physical activity or sedentary behavior.

A study conducted in Japan examined the validity of total EE (kcal) estimates with 12 activity monitors including the ASP and the AG compared with doubly labelled water method under free living conditions and with metabolic chamber methods under laboratory conditions [27]. Murakami et al. [27], found that the AG estimated, on average, less total EE (i.e., 524 kcal/day) compared with doubly labelled water method; however, no difference was found for total EE between the ASP and doubly labelled water method. Despite these differences, Murakami et al. [27] however, did not directly compare total EE estimated from ASP and AG under free living and laboratory conditions. In a study comparing the ASP and the AG in free living environments [8], differences in the mean of sedentary behavior time and sedentary breaks between ASP and AG with normal and low-frequency settings were found. Notably, the difference of measuring various intensities of physical activity between ASP and AG has not been examined yet.

Despite their potential differences, to our knowledge, there is yet no laboratory-based study comparing AG and ASP estimated EE for healthy adult physical activities and sedentary behaviors. Conducting laboratory-based comparison studies can provide key evidence on the differences between physical activity and sedentary behavior outcomes from different devices. Laboratory conditions can remove potential variability of estimates on physical activity and sedentary behavior that may be present while undertaking physical activity and sedentary behavior under free-living conditions. Laboratory settings also provide a more controlled experiment for estimating differences of specific physical activity and predicted EE. Information on differences in physical activity, sedentary behavior, and EE estimates between measurement devices is necessary for interpretation of previous studies and provides useful information for public health and sports scientists to choose the appropriate accelerometer relevant to their purposes.

The aim of this study was to estimate the agreement of EE (METs) for specific physical activities and sedentary behaviors between the ASP and AG accelerometers for common daily living activities under laboratory conditions.

## 2. Materials and Methods

### 2.1. Participants

Using snowball sampling, 30 undergraduate and graduate (undergraduate students: *n* = 16, graduate students: *n* = 14; mean ± standard deviation (SD) age = 23.8 ± 3.5 years, height = 167.8 ± 11.7 cm, weight = 60.5 ± 12.8 kg, body mass index = 21.4 ± 2.4 kg/m^2^) students from the faculty of sport sciences in Waseda university (Tokorozawa, Saitama, Japan) a university in Tokyo were recruited between December 2016 and May 2017 to participate in this study. The participants were 14 men (mean ± SD age = 24.9 ± 4.2 years, height = 175.9 ± 10.1 cm, weight = 70.0 ± 12.9 kg, body mass index = 22.5 ± 2.6 kg/m^2^) and 16 women (mean ± SD age = 22.8 ± 2.6 years, height = 160.0 ± 7.6 cm, weight = 52.1 ± 4.9 kg, body mass index = 20.4 ± 1.8 kg/m^2^). Significantly different characteristics in participants were found between men and women for height, weight, and body mass index (independent *t*-test, *t* = 2.6–5.1, df = 28, *p* < 0.05). Our sample size of 30 was sufficient to detect an effect size of d = 0.5, with a power of 75% and alpha of 0.05, and was consistent with previous similar studies [28,29]. We excluded participants who reported being unable to engage in any physical activities or those who reported health problems that might interfere with study tasks. Written informed consent was obtained from each participant on entry into the study. The Academic Research Ethical Review Committee at Waseda University, Japan (ID: 2016-047) approved the study.

### 2.2. Equipment

Participants simultaneously wore two accelerometers, including an AG (GT3X+) and an ASP (HJA-350IT) on their hip. Accelerometers were symmetrically attached to the left and right hip with an elastic belt to avoid contacts between two devices and fall accidents. The hip placement of accelerometers was randomized based on whether there the participants ID was an odd or even value.

### 2.3. Procedures

Participants were asked to conduct 22 activities in three categories (including eight sedentary, eight household, and six ambulatory activities; Table 1). Table 1 shows the compendium of physical activity [30], which has been widely accepted as METs intensity values in human physical activity. During the protocol, beginning and end times in each activity and group were managed by a research assistant using a timer that was synchronized with the accelerometers. All participants underwent the same procedure. Consistent with previous studies [23,25], each activity was selected based on representative or common activities of daily living undertaken by adults. Each activity was performed continuously for five minutes. Groups of activity followed the order outlined in Table 1. There were two minutes of rest between the groups of activities but no rest between each activity within the same category. Because there was no rest period between activities within each category, the first and last minute of data collection for each activity was excluded from the analysis. The ambulatory category consisted of walking on a treadmill (Jog forma; TECHNOGYM S.p.A, Cesena FC, Italy) at four different speeds (3.2, 4.2, 6.0, and 8.4 km/h), treadmill grade was held constant at 0%.

### 2.4. Data Management

ASP (HJA-350IT) has a proprietary algorithm with the ratio of un-filtered acceleration to filtered acceleration (ACCunfil/ACCfil), which estimates the METs of various activities. The algorithm using the ACCunfil/ACCfil ratio can categorize sedentary and non-locomotive or locomotive activity depending on the acceleration by using a multi-regression data processing system with high validation under laboratory conditions with a correct classification percentage of over 95% [23]. ASP has a weight of 60 g including battery and a size of 80 × 20 × 50 mm and can be attached to the waist. Unlike the AG accelerometer equations, the ASP estimates METs with no control and choice of setup for predicting METs, except for interval length of 10 or 60 s. Numerous studies examined physical activity and sedentary behavior outcomes from the AG and ASP data summarized in each 60-s intervals for adults [4,31,32,33]. In this study, 60-s intervals were used in the analysis. The ASP has a liquid crystal screen that shows the amount of activity (exercise per day), activity intensity (METs per minute), and steps (per day) in real-time. Settings for the ASP display can be modified to show various outcomes depending on the aim of the research. In this study, the ASP display was set to show physical activity time only. Data processing was conducted using the proprietary Omron health management software BI-LINK.

AG devices can measure activity counts, EE (kcal), activity intensity (METs), steps, and amount of time in various intensities (e.g., sedentary behavior and, light, moderate, and moderate-to-vigorous intensity of physical activity). In this study, AG measured activity intensity (METs). The AG can be worn on the hip or the wrist. AG accelerometer output is digitized by a 12-bit analogue-to-digital convertor at a rate of 30 to 100 Hz and recorded 1 to 60-s intervals. The AG in this study was initialized by setting raw acceleration signal at 30 Hz and recorded 60 s intervals. Data processing was conducted using the proprietary ActiLife^®^ software (version 6.10.4). In this study, AG output was measured using two types of equations; The uniaxial (VT) equation, and the triaxial (VM) equation. We estimated EE from the AG via four equations, Freedson [17] and Santos-Lozano VT [19] as a uniaxial setting (VT), Sasaki [18] and Santos-Lozano VM as a triaxial setting (VM) to estimate METs (Table 2) [17,18,19].

### 2.5. Statistical Analysis

METs predicted from the ASP and AG were presented as mean ± SD for each of the sedentary, household, and ambulatory activities by using each device equation (ASP, AG: Freedson, Sasaki, Santos-Lozano (VT and VM)). Paired *t*-tests compared mean METs for each activity between the ASP and the four AG equations, respectively. Bland and Altman analysis with 95% limits of agreement (LoA) assessed the bias and agreement between the ASP and each of the four AG equation estimates of METs for each activity. It is recommended that 95% of the data points should be within 1.96 SD of the LoA [34]. Relative to the two devices, positive mean difference suggests an overestimate and a negative mean difference suggests an underestimate of ASP estimated values compared to each activity estimated by the four AG equations. Statistical significance was reached if *p*-values were less than 0.05. IBM SPSS Statistics 22 software (IBM Japan Inc., Tokyo, Japan) was used to undertake the analysis.

## 3. Results

Thirty participants completed the study. Non-valid accelerometer data for specific activities were excluded due to premature termination of the activity prior to 5 min (brisk walking, *n* = 1; jogging, *n* = 1) and initial setting error (jogging, *n* = 1; up and down stairs, *n* = 1). The ASP and the AG using the Freedson and the Sasaki equations estimated all eight sedentary activities as ≤1.5 METs, while the AG using the Santos-Lozano VT and the Santos-Lozano VM equations measured all sedentary activities as above 1.5 METs (Table 3). Among the eight household activities, the ASP estimates ranged from 1.6 to 3.2 METs, except for standing, and sit-to-stand activities, which were measured as ≤1.5 METs. For the six ambulatory activities, the ASP estimates ranged from 3.0 to 9.9 METs; in contrast, four equations in the AG estimated slow-walking ranging from 2.4 to 2.9 METs as ≤2.9 METs. AG categorized light intensity of physical activity as 1.6 to 2.9 METs; even though other activities were estimated >3.0 METs, AG categorized moderate-to-vigorous physical activity as ≥3.0 METs. ASP and AGs measured mean METs in sedentary, household, and ambulatory activities within 1.5 METs differences from the compendium of physical activity, except for lifting, walking with bag and jogging (more than 1.71, 3.69, and 2.68 METs, respectively; data not shown in table).

The MET estimates for most activities using the Freedson, Sasaki, Santos-Lozano VT, and Santos-Lozano VM equations were significantly different (20, 20, 19, and 18 of 22 activities, respectively, *p* < 0.05; Table 4). A total of 14 out of 22 activities had significant differences of predicting METs between ASP and AG using four equations in common (seven were sedentary activities, i.e., laying, watching TV, sitting, reading, smart phones, PC (Internet) and PC (Work); five were household activities, i.e., standing, sit to stand, laundry, washing dishes, lifting; two were ambulatory activities, i.e., jogging, up and down stairs; *p* < 0.05; Table 4). There were significant differences in predicted METs values for most sedentary activities between the ASP and the AG using all the equations (*p* < 0.05), except for the Freedson equation (talking, *p* = 0.08). The Sasaki equation tended to estimate lower METs for most activities (mean difference range: sedentary. 0.2 to 0.6 METs; household, 0.2 to 0.9 METs; ambulatory, 0.1 to 1.7 METs; *p* < 0.05) compared to the ASP, while predicted METs values of household activities estimated for the ASP were significantly higher than the Freedson equation (mean difference = 0.1 to 1.2 METs, *p* < 0.01) except for the standing and sit to stand activities (mean difference = −0.2 to −0.3 METs, *p* < 0.01). The lowest differences in predicted METs data between the ASP and the AG occurred during the ambulatory activities. For ambulatory activities, relative to the ASP, the Freedson and the Sasaki equations significantly estimated lower METs for most activities compared to the ASP (mean difference range: 0.1 to 2.3 METs; *p* < 0.05 for all), except for climbing up and down stairs. Slow walking, brisk walking, and walking with bag had no significant differences between the ASP and each two AG equations (slow walking, Santos-Lozano VT and VM; brisk walking, Santos-Lozano VT and Sasaki; walking with bag. Freedson and Sasaki). Bland and Altman analysis indicated fixed bias between the ASP and the AG using all the equations in three activities (laying, watching TV, and standing), except for laying in the Sasaki equation and for standing in the Santos-Lozano VM equation. While the sedentary and household activities had fixed bias in almost all activities, the ambulatory activities had a better agreement between the ASP and AG than other activities (Figure 1).

## 4. Discussion

This is the first study comparing the ASP and AG for measuring intensity of EE (METs) in specific physical activities and sedentary behaviors under laboratory conditions. Overall, this study found low agreement of activity intensity estimates between ASP and AG. Additionally, the estimates of sedentary and household activities between the two devices had a systematic bias in almost all activities. Differences in outcome using these devices were larger in sedentary and household activities than ambulatory activities. Specifically, relative to the ASP the three equations of AG: the Freedson, Santos-Lozano VT, and Santos-Lozano VM equations mostly estimated rate of EE (METs) higher for sedentary activities and lower for ambulatory activities, whereas only the Sasaki equation tended to estimate lower for all activities. Also, MET values found for common activities in this study were consistent with MET values proposed by Ainsworth et al. [30] for similar activities, except for lifting, walking with bag, and jogging. With no previous studies, these findings provide important preliminary evidence on using these two accelerometer devices.

For outcomes of common activities for adults, our results for the ASP devices were similar with one study which provided the estimation model for intensities of physical activity in ASP [25]. A previous study looking at MET estimates that used the Douglas bag method as a gold standard found MET estimates of 12 specific activities of daily living (mean ± SD, METs); personal computer work (1.12 ± 0.02 METs), dishwashing (1.84 ± 0.34 METs), vacuuming (2.97 ± 0.52 METs), slow walking (3.3 km/h, 3.12 ± 0.45 METs), normal walking (4.2 km/h, 3.67 ± 0.55 METs), brisk walking (6.0 km/h, 4.70 ± 0.76 METs), and jogging (8.4 km/h, 9.42 ± 0.98 METs) [25]. Our ASP MET outcomes measuring common activities were concordant with these previous outcomes from the Douglas bag method. Our findings enhance the comparability of METS outcomes between ASP and AG devices which will help researchers to select the appropriate accelerometer devices [24].

Generally, while accelerometers worn on the hip have a good assessment of measuring activities in moderate-to-vigorous physical activity (i.e., normal walking, jogging), they may not be best suited to detect light intensity of physical activity (i.e., arm movements, standing, and sitting) [35]. Therefore, previous algorithms had the challenge of accurately distinguishing between light intensity of physical activity and sedentary behavior [36,37]. However, the current study showed that ASP provided more detail of different predicted METs in each sedentary and household activity compared to grouped categories using the AG. This study found that while ASP devices can distinguish sedentary and household activities, AG devices were not sensitive to these different activities. Current findings in the AG results are consistent with previous studies in younger adults [38] and older adults [39] which showed Freedson, Santos-Lozano VT, and Santos-Lozano VM equations could not distinguish different sedentary activities. In relation with sedentary activities, the four AG equations calculated the means of METs values almost in all sedentary activities as the lowest in initial setting (1.44, 0.68, 2.12, and 1.56 METs, respectively). AG has been previously reported to have low accuracy in measuring light intensity of physical activity, especially for sedentary and household activities [24]. Additionally, this study showed that all the tested AG equations estimated similar METs between household and sedentary activities. Conversely, ASP was sensitive to different movements in household and sedentary activities (mean of METs values ranged from 1.0 to 1.4 METs in sedentary activities, and 1.6 to 3.2 METs in household activities) except for static postures (e.g., standing and sit to stand). Considering ASP devices calculate 1.0 METs as the lowest METs in an initial setting when ASP measures no movement, the ASP can detect each activity in low intensity. There are two main reasons for the differences between ASP and AG. Firstly, there are differences in initial settings in measuring the lowest METs for ASP and the four equations in AG. Secondly, different algorithms are used for measuring physical activity and sedentary behavior, especially for low-intensity activities, in the ASP and AG devices [24,25]. Differences in household activities between ASP and AG devices with the four equations depended on the activities. The ASP may have better accuracy than the AG for distinguishing sedentary, household or locomotive activities using the ACCunfil/ACCfil algorithm [23] and AG may be unable to detect small movements like sedentary activities. AG algorithms were also not designed for estimating EE in light intensity of physical activity and sedentary behavior. In contrast, comparing to ambulatory activities we found that there were a smaller number of specific activities with the difference between the ASP and AG. Thus, agreements between AG and ASP for measuring walking activities in the ambulatory category are better than sedentary and household category. Three different speeds, ranging from slow to brisk walking on a treadmill (3.2, 4.2, and 6.0 km/h) had no significant differences between the ASP and each two AG equations, with a mean difference range from 0.02 to 0.57 METs. One study in Japan validated the measurement of step-count and EE during walking activity from three research-grade accelerometers, ASP, Lifecorder EX (Suzuken Co., Ltd., Nagoya, Japan) and Actimarker (Panasonic Electronic Works, Ltd., Osaka, Japan) under laboratory conditions with direct observations and using the Douglas bag method [40]. Park et al. [40] showed that ASP had a better accuracy of measuring EE (METs) of treadmill walking at three different speeds than uniaxial accelerometer Lifecorder EX, and had a high validity of measuring walking activities using the Douglas bag method [40]. However, as we did not directly compare to a gold standard of EE (e.g., doubly labelled water or Douglas bag methods) we cannot clarify which accelerometer (ASP or AG) had a better accuracy of measuring specific activities in this study.

Many previous studies have shown that triaxial data have more advantages in measuring physical activity and sedentary behavior than uniaxial data in accelerometer devices. Likewise, the axial setting can provide more accurate estimates of METs [1,7,41,42,43]. In this study, the triaxial accelerometer ASP and the AG using two uniaxial equations (Freedson and Santos-Lozano VT) and two triaxial equations (Sasaki and Santos-Lozano VM) were used. Our findings using the AG accelerometer data from VM were not different for the ASP measuring METs of sedentary, household, and ambulatory activities from only using the AG accelerometer data from the VT. In this study, we used four equation models for predicting EE (METs) from accelerometer raw data (count) for adults; nevertheless, AG has several available options for accelerometer raw data analysis. For instance, a machine learning method is an effective analysis using triaxial raw data to predict EE and classify physical activity intensities more accurately than traditional methods which are used to convert the accelerometer raw data (count) to EE [44,45]. Therefore, different options for data analysis may cause different results from the present outcome. Thus, it is important to investigate the difference between ASP and AG using different analyses in future studies.

This study has limitations. This study cannot identify which accelerometers or estimation equations were more accurate for EE because there are no indirect calorimeters (i.e., portable metabolic units, or room calorimeter) or doubly labelled water method as criterion methods [46]. However, this study used the most commonly used accelerometer (AG) in field-based research, instead of the gold standard measure of EE because the primary purpose of this study was to assess the difference in the rate of EE estimated between two well-validated hip worn accelerometers, the ASP [8,25] and AG [47,48]. Therefore, we could clarify the differences in common activities between ASP and AG. These findings may help researchers to select research-grade accelerometers for research measuring physical activity and sedentary behavior. Second, the current study had a small sample size and was composed of young adults. Participants were recruited from only university students, which limits generalizability to the broader adult population.

## 5. Conclusions

The present study demonstrated that there were measurement differences between the ASP and AG accelerometers in measuring physical activity and sedentary behavior. Additionally, none of the AG equations had good agreements with the ASP device. Differences between the ASP and AG were larger in sedentary and household activities. This study provides preliminary evidence on differences between AG and ASP devices in measuring physical activity and sedentary behavior. Such evidence is informative for public health and sport science researchers and clinicians in applying these accelerometer devices and interpreting previous studies which used these devices.

## Figures and Tables

**Figure 1 ijerph-16-03126-f001:**
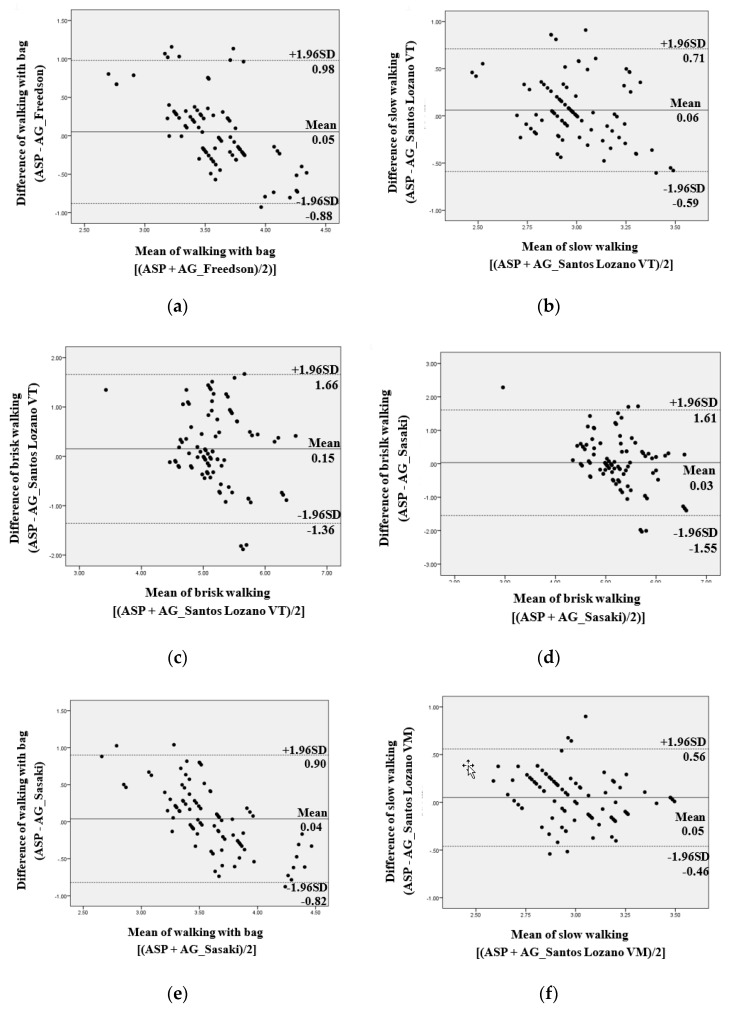
Bland and Altman analysis: Active Style Pro and ActiGraph using four equations (METs). (**a**) Freedson, walking with bag. (**b**) Santos-Lozano VT, slow walking. (**c**) Santos-Lozano VT, brisk walking. (**d**) Sasaki, brisk walking. (**e**) Sasaki, walking with bag. (**f**) Santos-Lozano VM, slow walking.

**Table 1 ijerph-16-03126-t001:** Each category activities performed in this study.

Category	Activity	Description of Activity	Compendium METs (Code)
Sedentary (eight activities)	Lying down	Lying in supine position awake. Avoid bodily movement.	1.3 (07011)
	Watching television	Lying in lateral position awake.	1.0 (07010)
	Sitting	Seated, quietly, Avoid bodily movement.	1.2 (07021)
	Reading magazine	Seated, quietly, reading a magazine	1.3 (09030)
	Talking	Seated, free talking	1.5 (09055)
	Using smartphone	Seated, using a smartphone	1.5 (09055)
	Searching Internet	Seated, using a laptop	1.3 (09040)
	Desk work	Seated, typewriting using a personal computer with sitting in a chair	1.3 (11770)
Household (eight activities)	Standing	Standing on the floor. Avoid bodily movement.	1.2 (07040)
	Sit to Stand	Sit to stand each 30 sec	2.8 (02024)
	Arm moving	Standing/ambulating, write on the whiteboard	1.8 (09020)
	Sweeping	Standing/ambulating, window cleaning (Height 180 cm)	3.2 (05022)
	Laundry	Carrying clothes from a laundry basket and hanging up clothes	2.3 (05095)
	Dishwashing	Washing the dishes	2.3 (05041)
	Moving a small load	Lifting a small load of 5 kg and unloading it after a few steps	5.0 (05121)
	Vacuuming	Vacuuming clean in a room while moving	3.3 (05043)
Ambulatory (six activities)	Slow walking	Walking on a treadmill 3.2 km/h	2.8 (17152)
	Normal walking	Walking on a treadmill, 4.2 km/h	3.0 (17180)
	Brisk walking	Walking on a treadmill, 6.0 km/h	4.3 (17200)
	Walking with a backpack	Walking on a treadmill 3.2 km/h with a 3 kg backpack	7.0 (17010)
	Jogging	Jogging on a treadmill 8.4 km/h	9.0 (12040)
	Climbing up and down stairs	Walking up and down stairs at a self-selected speed	5.0 (17026)

Note: treadmill using the jog forma (TECHNOGYM S.p.A, Cesena FC, Italy). Treadmill setting: no grade (0%). METs: metabolic equivalents. Compendium METs values adapted from Ainsworth et al. [30].

**Table 2 ijerph-16-03126-t002:** Prediction equations used in ActiLife^®^ to obtain METs from the ActiGraph (AG) accelerometer.

Monitor	The Equation Used to Calculate METs
AG (Freedson)	1.439008 + 0.000795 × VT count/minute
AG (Santos-Lozano VT)	2.118089 + 0.000662 × VT count/minute
AG (Sasaki)	0.668876 + 0.000863 × VM count/minute
AG (Santos-Lozano VM)	1.546618 + 0.000658 × VM count/minute

Note: METs: metabolic equivalents, VT: vertical axis, VM: vector magnitude.

**Table 3 ijerph-16-03126-t003:** Activity-specific predicted METs from each prediction equation in AG and ASP.

Activity		ASP	AG (Freedson)	AG (Santos-Lozano VT)	AG (Sasaki)	AG (Santos-Lozano VM)
Sedentary activities	Laying	1.01 ± 0.03	1.44 ± 0.02	2.12 ± 0.02	0.68 ± 0.09	1.56 ± 0.07
	Watching TV	1.02 ± 0.06	1.46 ± 0.09	2.14 ± 0.07	0.73 ± 0.23	1.59 ± 0.18
	Sitting	1.05 ± 0.11	1.45 ± 0.03	2.12 ± 0.03	0.69 ± 0.10	1.57 ± 0.07
	Reading	1.09 ± 0.13	1.44 ± 0.00	2.12 ± 0.00	0.69 ± 0.04	1.56 ± 0.03
	Talking	1.41 ± 0.20	1.44 ± 0.01	2.12 ± 0.01	0.73 ± 0.12	1.59 ± 0.09
	Smart phone	1.12 ± 0.17	1.44 ± 0.00	2.12 ± 0.00	0.68 ± 0.04	1.56 ± 0.03
	PC (Internet)	1.08 ± 0.12	1.44 ± 0.01	2.12 ± 0.00	0.69 ± 0.05	1.56 ± 0.04
	PC (Work)	1.12 ± 0.13	1.44 ± 0.00	2.12 ± 0.00	0.68 ± 0.04	1.55 ± 0.03
Household activities	Standing	1.13 ± 0.19	1.44 ± 0.01	2.12 ± 0.01	0.69 ± 0.05	1.56 ± 0.04
	Sit to stand	1.31 ± 0.18	1.53 ± 0.06	2.19 ± 0.05	1.04 ± 0.12	1.83 ± 0.09
	Arm moving	1.65 ± 0.20	1.46 ± 0.02	2.13 ± 0.02	0.85 ± 0.12	1.68 ± 0.09
	Windows	2.16 ± 0.33	1.47 ± 0.03	2.14 ± 0.03	1.18 ± 0.30	1.93 ± 0.23
	Laundry	2.16 ± 0.35	1.86 ± 0.36	2.47 ± 0.30	1.88 ± 0.67	2.47 ± 0.51
	Washing dishes	1.71 ± 0.23	1.45 ± 0.05	2.13 ± 0.04	0.83 ± 0.18	1.67 ± 0.14
	Lifting	3.29 ± 0.84	2.02 ± 0.34	2.60 ± 0.28	2.62 ± 0.80	3.03 ± 0.61
	Vacuuming	2.67 ± 0.40	1.66 ± 0.24	2.30 ± 0.20	2.26 ± 0.84	2.76 ± 0.64
Ambulatory activities	Slow Walking	3.02 ± 0.21	2.45 ± 0.36	2.96 ± 0.30	2.54 ± 0.35	2.97 ± 0.27
	Normal walking	3.67 ± 0.25	3.45 ± 0.50	3.80 ± 0.42	3.48 ± 0.50	3.69 ± 0.38
	Brisk walking	5.28 ± 0.54	5.05 ± 0.82	5.13 ± 0.68	5.24 ± 0.84	5.03 ± 0.64
	Walking with bag	3.61 ± 0.25	3.56 ± 0.51	3.88 ± 0.43	3.57 ± 0.54	3.76 ± 0.41
	Jogging	9.91 ± 0.98	7.90 ± 1.40	7.50 ± 1.17	8.13 ± 1.58	7.23 ± 1.21
	Up and down stairs	4.18 ± 0.71	4.93 ± 0.69	5.02 ± 0.57	4.99 ± 0.73	4.84 ± 0.55

Note: Data shown as mean ± standard deviation, METs: metabolic equivalents, VT: vertical axis, VM: vector magnitude.

**Table 4 ijerph-16-03126-t004:** Differences and agreement for each activity (METs) between ASP compared with AG using four equations.

	Freedson			Santos-Lozano VT			Sasaki			Santos-Lozano VM		
	Difference	LoA		Difference	LoA		Difference	LoA		Difference	LoA	
		High	Low		High	Low		High	Low		High	Low
Sedentary activities												
Laying	−0.44 **	−0.37	−0.51	−1.12 **	−1.05	−1.19	0.32 **	0.50	0.14	−0.55 **	−0.40	−0.70
Watching TV	0.44 **	−0.23	−0.65	−1.12 **	−0.93	−1.31	0.29 **	0.76	−0.18	−0.57 **	−0.20	−0.94
Sitting	−0.39 **	−0.16	−0.62	−1.07 **	−0.84	−1.30	0.36 **	0.64	0.08	−0.51 **	−0.26	−0.76
Reading	−0.35 **	−0.1	−0.6	−1.03 **	−0.78	−1.28	0.41 **	0.65	0.17	−0.47 **	−0.23	−0.71
Talking	−0.04	0.34	−0.42	−0.71 **	−0.33	−1.09	0.68 **	1.00	0.36	−0.18 **	0.14	−0.5
Smart phone	−0.32 **	0.01	−0.65	−1.00 **	−0.67	−1.33	0.44 **	0.75	0.13	−0.44 **	−0.13	−0.75
PC (Internet)	−0.36 **	−0.13	−0.59	−1.04 **	−0.81	−1.27	0.39 **	0.63	0.15	−0.48 **	−0.25	−0.71
PC (Work)	−0.32 **	−0.07	−0.57	−1.00 **	−0.75	−1.25	0.44 **	0.69	0.19	−0.44 **	−0.19	−0.69
Household activities												
Standing	−0.31 **	0.07	−0.69	−0.99 **	−0.61	−1.37	0.44 **	0.82	0.06	−0.43 **	−0.05	−0.81
Sit to stand	−0.21 **	0.16	−0.58	−0.88 **	−0.52	−1.24	0.27 **	0.58	−0.04	−0.52 **	−0.21	−0.83
Arm moving	0.19 **	0.58	−0.2	−0.49 **	−0.10	−0.88	0.80 **	1.13	0.47	−0.04	0.30	−0.38
Windows	0.70 **	1.35	0.05	0.02	0.67	−0.63	0.99 **	1.50	0.48	0.23 **	0.72	−0.26
Laundry	0.30 **	0.92	−0.32	−0.31 **	0.27	−0.89	0.28 **	1.24	−0.68	−0.31 **	0.40	−1.02
Washing dishes	0.26 **	0.70	−0.18	−0.42 **	0.02	−0.86	0.88 **	1.29	0.47	0.04 *	0.43	−0.35
Lifting	1.27 **	2.62	−0.08	0.69 **	2.08	−0.7	0.67 **	1.60	−0.26	0.26 **	1.19	−0.67
Vacuuming	1.01 **	1.67	0.35	0.37 **	1.03	−0.29	0.41 **	1.56	−0.74	−0.09	0.72	−0.90
Ambulatory activities												
Slow walking	0.57 **	1.32	−0.18	0.06	0.71	−0.59	0.49 **	1.12	−0.14	0.05	0.56	−0.46
Normal walking	0.21 **	1.26	−0.84	−0.13 **	0.77	−1.03	0.19 **	1.09	−0.71	−0.02	0.69	−0.73
Brisk walking	0.23 *	1.95	−1.49	0.15	1.66	−1.36	0.03	1.61	−1.55	0.24 *	1.53	−1.05
Walking with bag	0.05	0.98	−0.88	−0.27 **	0.52	−1.06	0.04	0.90	−0.82	−0.15 **	0.50	−0.80
Jogging	2.32 **	5.01	−0.37	2.41 **	5.14	0.30	1.78 **	4.90	−0.72	2.68 **	5.34	0.62
Up and down stairs	−0.75 **	0.76	−2.26	−0.84 **	0.56	−2.24	−3.93 **	0.79	−2.41	−0.66 **	0.77	−2.09

Note: Positive means of difference showed an overestimate and the negative mean difference showed an underestimate of ASP estimated values compared to four AG equations estimated each activity. METs: metabolic equivalents, VT: vertical axis, VM: vector magnitude, LoA: limits of agreement. The upper limits of the agreement (high): mean difference + 1.96 standard deviation of the different scores. The limits of the agreement (low): Mean difference −1.96 standard deviations of the different scores. * *p* < 0.05, ** *p* < 0.01.

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
