# Peer review of "Physical Activity and Sedentary Behavior Assessment: A Laboratory-Based Evaluation of Agreement between Commonly Used ActiGraph and Omron Accelerometers"

_ijerph, 2019, doi:10.3390/ijerph16173126_

Round 1

Reviewer 1 Report

I was honored to review the manuscript entitled "Physical activity and sedentary behavior  assessment: A laboratory-based evaluation of agreement between commonly used ActiGraph and  Omron accelerometers" submitted to International Journal of Environmental Research and Public Health.

The aim of this work was to estimate the agreement of EE for specific PAs and SBs between the ASP and AG accelerometers for common daily living activities under laboratory conditions.

Taking into account the multiple studies ongoing in the field this type of study is needed.  I have only few small remarks that authors should adress properly.

I recommend to accept the manuscript after minor revision.

Points that need correction:

 - please provide the list of abbreviations.

 - please provide the number of ethical approval

 - Introduction and Discussion section needs improvement

I recommend to accept the manuscript after minor revision.

Reviewer 2 Report

Given the increasing popularity of fitness/activity trackers and users’ beliefs that they provide accurate feedback, it is obviously essential to have empirical evidence about how effective such devices are in providing reliable evidence about the users’ activity.

The current study compared two different accelerometers to establish whether they provided similar data for users carrying out 22 similar activities in three categories from less to more strenuous. The study is timely in providing some evidence that questions the consistency of the 2 different methods of measurement used in the 2 devices. The findings of the study were that the two accelerometers did not provide similar data across a range of activities.

Materials and Methods

P5 line 192 “Mean ± standard deviation (SD) METs estimated from the ASP and AG were predicted” – is predicted the right word here – is it not measured (according to the algorithms used by the different devices)?

P 4 line 144: Are there any problems with participants wearing both devices at the same time? In many ways it’s a very neat aspect of the design but could there be any interference between the devices for example?

P6 line 213: not sure why this information about participants was included in the results section? “Thirty participants completed the study, including 16 men and 14 women. Mean ± SD age 213 was 23.8 ± 3.5 years, height was 167.8 ± 11.7 cm, weight was 60.5 ± 12.8 kg, and body mass index 214 (BMI) was 21.4 ± 2.4 kg/m².”

It would be useful to include weight & height split by gender. (It would even have been interesting to see gender differences in data, but I appreciate that would add an extra dimension!)

As someone who is not very familiar with this literature I found the extensive use of acronyms throughout confusing and annoying. I had to keep translating from the acronym to the words. The authors should reduce the acronyms and for example use the full terms for everyday rather than technical terms (e. g. physical activity instead of PA and sedentary behavior instead of SB). Also the author should define all acronyms on first use - p5 181 What are (e.g., SB, LPA, MPA, AND MVPA)?

Results

The descriptive data was presented clearly and the stats described clearly, although as well as reporting the Bland and Altman analysis results it would have been useful to have more interpretation of what the results mean. This was not referred to in the discussion.

P7 235: “The MET estimates for most activities using the Freedson, Sasaki, Santos-Lozano VT, and 235 Santos-Lozano VM equations were significantly different (20, 20, 19, and 18 of 22 activities, p < 236 0.05; table 4)” Make it explicit what the values were different from, i. e. from the ASP values (not each other).

If “Our ASP MET outcomes measuring common activities were concordance with these previous outcomes from Douglas bag method.” Are the authors suggesting that they would prefer to go with the ASP method than the AG.

Verbal Expression

Typos

P3 line 101: “should be should the researcher require the wearer be blinded from their activity levels” should read “should the researcher require the wearer be blinded from their activity levels”

P4 line 136/137 – odd expressional error –“This study had a sample size that was”

Page 14 line 314 “lower” should be “lower for”?

Page 14 , 327: “concordance with these” should be “concordant with these”

Table 7 Standing not sanding?
